# Remediation of heavy metal contaminated soil by asymmetrical alternating current electrochemistry

Jinwei Xu[1], Chong Liu[1], Po-Chun Hsu[2], Jie Zhao[1], Tong Wu[1], Jing Tang[1], Kai Liu[1] & Yi Cui [1,3]

Soil contamination by heavy metals constitutes an important environmental problem, whereas field applicability of existing remediation technologies has encountered numerous obstacles, such as long operation time, high chemical cost, large energy consumption, secondary pollution, and soil degradation. Here we report the design and demonstration of a remediation method based on a concept of asymmetrical alternating current electrochemistry that achieves high degrees of contaminant removal for different heavy metals (copper, lead, cadmium) at different initial concentrations (from 100 to 10,000 ppm), all reaching corresponding regulation levels for residential scenario after rational treatment time (from 30 min to 6 h). No excessive nutrient loss in treated soil is observed and no secondary toxic product is produced. Long-term experiment and plant assay show the high sustainability of the method and its feasibility for agricultural use.

[1] Department of Materials Science and Engineering, Stanford University, Stanford, CA 94305, USA. [2] Department of Mechanical Engineering, Stanford University, Stanford, CA 94305, USA. [3] Stanford Institute for Materials and Energy Sciences, SLAC National Accelerator Laboratory, 2575 Sand Hill Road, Menlo Park, CA 94025, USA. Correspondence and requests for materials should be addressed to Y.C. (email: yicui@stanford.edu)

With the increasing population and demand for agricultural land[1,2], soil contamination is becoming a serious global environmental crisis[3,4]. Heavy metals in soil resulting from anthropogenic activities is one of the most critical issues, particularly given a large number of widespread poisoning incidents[5–8]. Considering the high accumulation rate of heavy metals in the earth's upper crust due to the jumping global mining production and industrial demand[9,10], economical and efficient remediation of contaminated urban and agricultural land is in pressing need for a sustainable development prospect. In general cases, heavy metals in soil are in the form of cations and retained on soil particles by electrostatic attraction or forming chemical bonds with organic or inorganic ligand ions[11]. One remediation solution is soil washing with strong chelating agent[12], which liberates heavy metal cations from the functional groups on the surface of soil particles. However, there are three concerns hindering the application of this technology: the high consumption of chelating agents[13], the lack of efficient strategy to treat the washing effluent[14], and the excessive soil nutrient loss after washing[15]. Another idea is to use high-surface-area sorbent to decrease the mobility and bioavailability of heavy metal cations[16], but the slow capture speed and low capacity due to its physicochemical adsorption nature are the major drawbacks[17]. The stability of immobilized heavy metals also requires long-term monitoring[18]. Phytoremediation has been developed in recent years as a high energy efficient method[19]. Nevertheless, the extremely long treatment time makes it only suited for remote areas, and the heavy metals accumulated in biomass may cause secondary pollution.

The above technologies all extract or aggregate heavy metal cations with their oxidation states unchanged. However, the most compact and immobilized states of heavy metals are solid forms as charge-neutral metals or metal alloys. Therefore, the ultimate remediation goal is not only to separate heavy metal cations from soil matrix but also to reduce them to zero-valent metallic states, which not only enhances the remediation capacity but also provides the opportunity of heavy metal recovery. Electrochemistry is the handiest method for the reduction of heavy metal cations, and can also differentiate heavy metals from nutrient elements according to their reduction potentials. The current state-of-the-art electrochemical remediation method applies a direct current (DC) electric field to the soil to transport heavy metal species by electroosmosis and electrodeposits them to metallic states on the negative electrode[20]. However, field application of this technology is limited by the high DC voltage required to maintain a strong electric field (~100 V/m) for electroosmosis[21], the low ion migration speed in soil[22] and the large energy losses associated with water splitting at electrodes[23].

In this work, we show a remediation method composed of a recirculating soil washing system and an electrochemical filtration device, which achieves high degrees of heavy metal removal from contaminated soil under a range of different concentrations. This remediation method is based on a key concept of asymmetrical alternating current electrochemistry (AACE) to be explained later, which enables recycling of soil washing chemicals and eliminates secondary pollution. In addition, we synthesize amidoxime-functionalized electrodes to facilitate the electrodeposition process. We also provide insight into the transformation of heavy metals, which are reduced to their zero-valent metallic states. Finally, plant assay shows negligible soil degradation after the treatment. Our results are expected to serve as a tool for the recovery of heavy metals from the waste streams in various manufacturing and chemical industries.

## Results

**AACE remediation method.** The construction of our AACE remediation method (Fig. 1a) involves a recirculating chelating agent washing system and an AACE filtration device. The soil is excavated from contaminated sites to a treatment cylinder, with two ethylenediaminetetraacetic acid (EDTA) solution reservoir tanks attached on each side. A water pump circulates the EDTA solution to wash through the contaminated soil column. The soil-sorbed heavy metal cations are mobilized by forming heavy metal-EDTA complex and transported to the AACE filter, which is connected to an alternating current (AC) power supply. Figure 1b shows an illustration of the AACE filter, composed of two parallel amidoxime-functionalized porous carbon (Ami-PC) electrodes and a separator. After the AACE filtration, heavy metal cations are liberated from their chelation complex and electrodeposited to metallic states on the working electrode, and the EDTA solution is recycled for repeated use. The Ami-PC electrode was fabricated by coating a carbon felt with a precursor slurry of polyacrylonitrile (PAN) and activated carbon, followed by a hydrothermal reaction to substitute the nitrile functional groups in PAN with amidoxime functional groups[24] (Supplementary Fig. 1). The amidoxime has two functions: to modify the carbon felt surface to hydrophilic thus to fully utilize the high-surface area of the electrodes and, more importantly, to provide strong chelation sites (Supplementary Fig. 2) that can compete with EDTA to bind heavy metal cations and hence promote the electrodeposition efficiency. The nano-size activated carbon (~40 nm) serves to enhance the electrical conductivity of the amidoxime polymer. The scanning electron microscope (SEM) image in Supplementary Fig. 3a shows the morphology of the Ami-PC electrodes, with a pore size ranging from tens to hundreds of micrometers and a fiber diameter of ~20 µm. The magnified SEM image in Supplementary Fig. 3b shows the homogeneous amidoxime coating on carbon fibers.

For the washing effluent, calculation using Visual MINTEQ[25] in Supplementary Fig. 4 shows that ~100% extracted heavy metal cations occur as anionic complex (MEDTA$^{2-}$). If a DC voltage were applied, the negative charge of MEDTA$^{2-}$ would reject the negative electrode due to Coulomb repulsion, with limited heavy metal cations electrodeposited and a great energy loss in water splitting. To address this issue, we developed a new method that applies an asymmetrical alternating voltage to the Ami-PC electrodes (Fig. 1c). The working electrode was alternating between 5 and −10 V with durations of 0.5 and 2 ms, respectively, and the counter electrode was connected to ground. The process of the AACE method is explained in three steps in the schematics. In step I, all the ions are randomly distributed in the washing effluent. In step II, a 5 V bias is applied, and ions start to migrate and establish an electrical double layer on the surface of the Ami-PC electrode, with anions in the inner layer. The chelation sites of amidoxime will compete with EDTA to bind heavy metal cations thus stabilize the MEDTA$^{2-}$ on the electrode surface. In step III, the bias is reversed to −10 V, electrochemically reducing heavy metal cations to zero-valent particles. EDTA anions will lose their affinity for these charge-neutral particles and be repelled by the negative bias. During the soil washing process, some soil nutrient cations, such as calcium (Ca$^{2+}$) and magnesium (Mg$^{2+}$), can also be extracted and form chelation complex (NEDTA$^{2-}$) like heavy metals. Same as MEDTA$^{2-}$, NEDTA$^{2-}$ has negative charges and can be adsorbed to the electrode surface in step II. However, these nutrient cations do not undergo electrodeposition in step III because of their lower reduction potential, thus remain in the recycled EDTA solution and are given back to the soil matrix by the recirculating soil washing, which prevents future soil nutrient loss.

**Remediation performance investigation.** To quantitatively evaluate the remediation performance of the AACE method, a

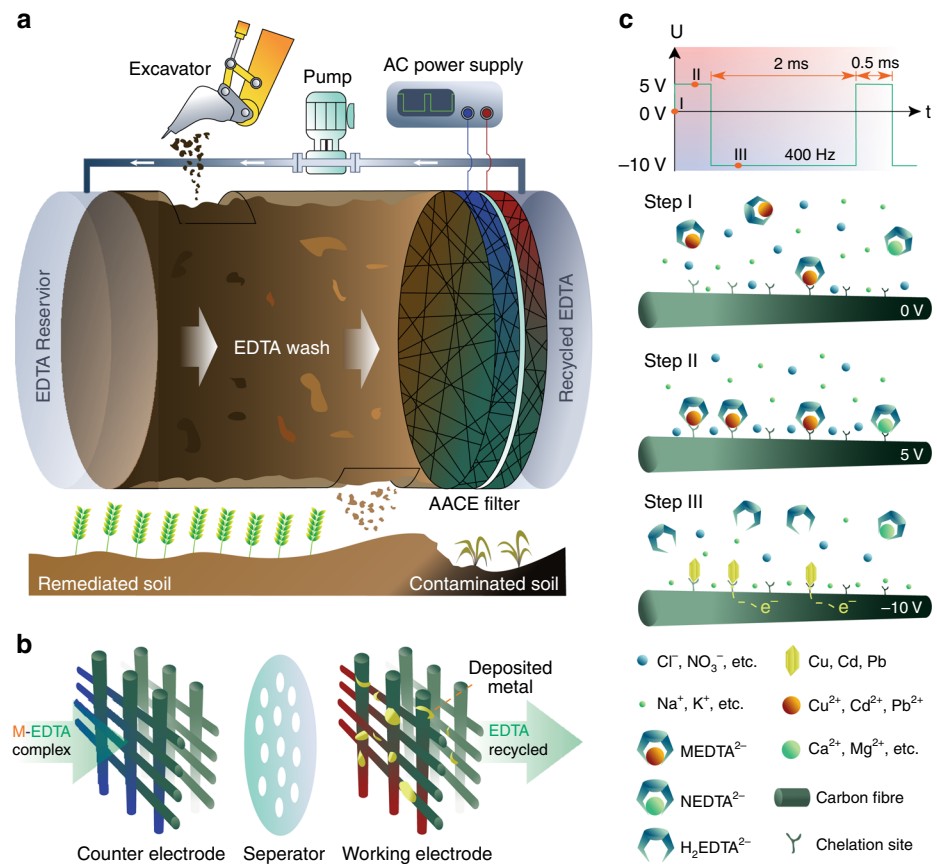

**Fig. 1** Working principle of the AACE method. **a** Schematic of an AACE remediation system. EDTA solution is recirculated to wash the contaminated soil. The AACE filter recovers heavy metal cations from the washing effluent and recycles the EDTA solution for repeated use. **b** Illustration drawing of the AACE filter, composed of two parallel Ami-PC electrodes and a separator. **c** The waveform of the applied bias and the physical process in the AACE filtration. In step I, all the ions are randomly dispersed in the washing effluent. In step II, a bias of 5 V applied, an electrical double layer is established on the surface of the Ami-PC electrode, with anions in the inner layer. The chelation sites will bind heavy metal cations and stabilize the $MEDTA^{2-}$ on the electrode surface. In step III, a bias of −10 V applied, heavy metal cations are electrochemically reduced to zero-valent particles. EDTA anions are repelled by the negative bias due to less affinity with the charge-neutral particles. Soil nutrient elements are reserved in this process because of their lower reduction potential

series of experiments using synthetically contaminated soil were conducted. The soil used in this study was collected from the O'Donohue Family Stanford Educational Farm. The soil was then air-dried at 70 °C, and passed through a 2-mm sieve to remove coarse fragments. Contaminated soil samples with different heavy metal (Cu, Pb, Cd) concentrations were synthetically prepared by thoroughly mixing the clean soil samples with the corresponding nitrate-salt solutions. After the spiking process, the contaminated soil samples were air-dried and aged for 1 month at 80 °C, in order to simulate practical polluted soil by decreasing the solubility and mobility of the heavy metals (Supplementary Fig. 5). Characteristics of the fresh and the aged soil samples, including soil texture, organic carbon, pH, and cation exchange capacity are provided in Supplementary Table 1, corroborating that the aging process didn't change the soil properties from that in field.

Considering the large variation in hazardous level among different contaminated sites[26] and the disparate safety standards for different heavy metals[27], three synthetically contaminated soil samples were prepared by spiking with 10,000 ppm Cu, 1000 ppm Pb, and 100 ppm Cd, respectively. The setting of their content is according to their toxicity and their typical concentrations found in contaminated sites. The heavy metal concentrations in these three soil samples during remediation treatments are shown in Fig. 2a–c. The AACE method successfully reduced the

concentrations of Cu, Pb, and Cd in the contaminated soil samples to below their California Human Health Screening Levels (CHHSL)[28] for residential scenario, which is similar compared with using fresh EDTA solution to wash the spiked soils. In addition, the AACE method recycled the EDTA solution for repeated use and therefore consumed very limited EDTA. For comparison, experiments with no bias on the electrodes failed to extract heavy metals from soil after the first soil washing cycle, because the EDTA solution had been saturated with heavy metal cations. Figure 2d shows that the ability of the AACE filter to recover heavy metals from washing effluent (defined as filtration efficiency) can be promoted by allowing a slower soil washing flow rate. The ultimate flow rate should be determined according to different contamination conditions: a too-high flow rate yields a poor filtration efficiency and a large amount of heavy metals remain in the recycled EDTA solution, while a too-low flow rate makes the remediation process slow and leads to more side reactions, hence wastes electricity energy. Therefore, the highest flow rate giving a filtration efficiency above 90% was chosen for the corresponding soil sample, as star marked in Fig. 2d. The concentration of magnesium in soil was also monitored during three different treatment methods at a flow rate of 0.1 ml min⁻¹ (Fig. 2e). For nutrient metal ions (Na⁺, Mg²⁺, etc.), they can also be washed out by the EDTA solution. However, they cannot be

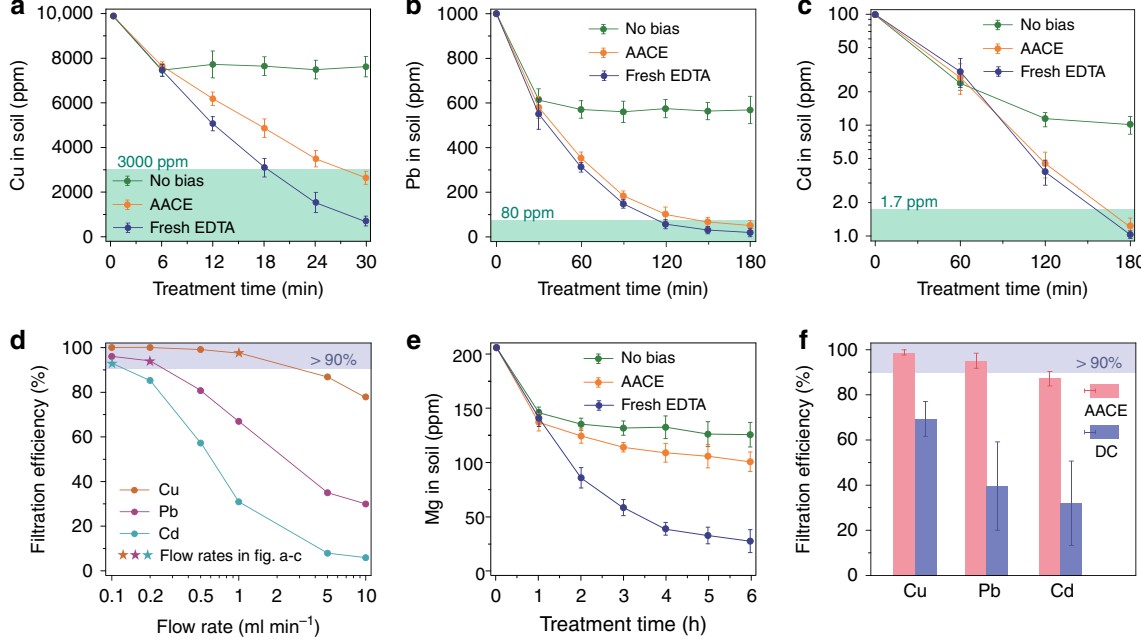

**Fig. 2** Remediation performance of the AACE method. **a**–**c** Comparison between the AACE remediation and two other methods on treating three soil samples spiked with 10,000 ppm Cu (**a**), 1000 ppm Pb (**b**), and 100 ppm Cd (**c**), respectively. 'No bias', using the same experiment set up as the AACE method and no bias is applied to the electrodes. 'Fresh EDTA', using fresh EDTA solution to wash the spiked soil without recirculation. Green region represents the CHHSL for residential scenario. **d** AACE filtration efficiency (the percentage of the heavy metal cations electrodeposited by the AACE filter from the washing effluent) as the function of soil washing flow rate. The highest flow rate providing a filtration efficiency above 90% was chosen for the corresponding treatment. **e** Concentration of Mg in soil under three different treatment methods. Flow rate, 0.1 ml min$^{-1}$. **f** Comparison of the filtration efficiency of the AACE method and the DC method. 'DC', using a −10 V DC for the electrochemical filtration. Flow rate, 0.1 ml min$^{-1}$. Error bars, mean ± s.d. ($n = 3$)

extracted out by the AACE filter because of their lower standard reduction potential. After the first washing cycle, the nutrient metal concentration in the EDTA solution established an equilibrium with the nutrient metal concentration in the soil. Consequently, when we used the recycled EDTA solution to wash the soil in the following cycles, it didn't wash out more nutrient metal ions. However, excessive nutrient loss happened when using fresh EDTA to wash the soil. To demonstrate the capability of the AACE method to treat mixed contamination, a soil sample simultaneously spiked with 10,000 ppm Cu, 1000 ppm Pb, and 100 ppm Cd was prepared. After 6 h of AACE treatment at a flow rate of 0.1 ml min$^{-1}$, the concentrations of Cu, Pb, and Cd in soil were reduced to 2874, 47, and 1.2 ppm, respectively, all below the CHHSL (Supplementary Fig. 6). Experiment using a −10 V DC instead of the asymmetrical AC was carried out to treat the mixture contaminated soil. Figure 2f shows that, at the same flow rate of 0.1 ml min$^{-1}$, the DC method can only extract heavy metals from the washing effluent by 30–70%, which is much lower than the AACE method.

**Remediation mechanism study.** The heavy metal species extracted by the AACE method and the DC method were further characterized and compared with each other to investigate the remediation mechanism. A synthetically contaminated soil sample spiked with an equivalent quantity of Cu, Pb, and Cd (1000 ppm each) was prepared for the investigation. After six cycles of soil washing (3 h at a soil washing flow rate of 0.2 ml min$^{-1}$), the morphologies of the heavy metal species on the Ami-PC electrode were characterized by SEM (Fig. 3a, c). In the AACE method, there are many micrometer-sized heavy metal particles attached

onto the Ami-PC electrode. In contrast, the Ami-PC electrode with DC bias was covered with a uniform thick shell, showing no crystal structure. The electrodes were further characterized by energy-dispersive X-ray spectroscopy (EDS) (Fig. 3b, d). Strong peaks of Cu, Pb, and Cd can be found for the AACE case, while the DC method accumulated a large amount of Ca element on the electrode. Considering the low electrochemical reduction potential of Ca$^{2+}$ and the low solubility of Ca(OH)$_2$, the DC extraction is primarily due to precipitation at the high pH zone resulted from hydrogen generation near the negative electrode. To check this hypothesis, X-ray photoelectron spectroscopy (XPS) was performed to determine the chemical state of the heavy metal species extracted by the DC method and the AACE method (Fig. 3e). In the DC case, the strong satellite peaks in the Cu 2p spectrum confirmed the copper species as Cu(II). While for the AACE method, the XPS peaks were observed at 932.8 and 952.6 eV with weak shaken-up structure indicating that most of the copper species existed as metallic state. For the Pb 4f and Cd 3d spectra, observed peaks of the Pb and Cd species extracted by the DC method were coincident with that of their metal hydroxides. As for the AACE method, peak separation analysis suggested that the extracted Pb and Cd species were a mixture of zero-valent state and divalent state with a majority existing as metal. Therefore, the AACE method successfully recovered the heavy metal cations to the zero-valent particles attached on the electrode. However, for the DC method, most of the heavy metal cations were precipitated with the hydroxyl ions produced by the negative bias on the electrode. This precipitation mechanism failed to reduce the heavy metal cations to zero-valent states and many soil nutrient cations like Ca$^{2+}$ were also extracted from the washing effluent. Unlike metals or metal alloys, these metal

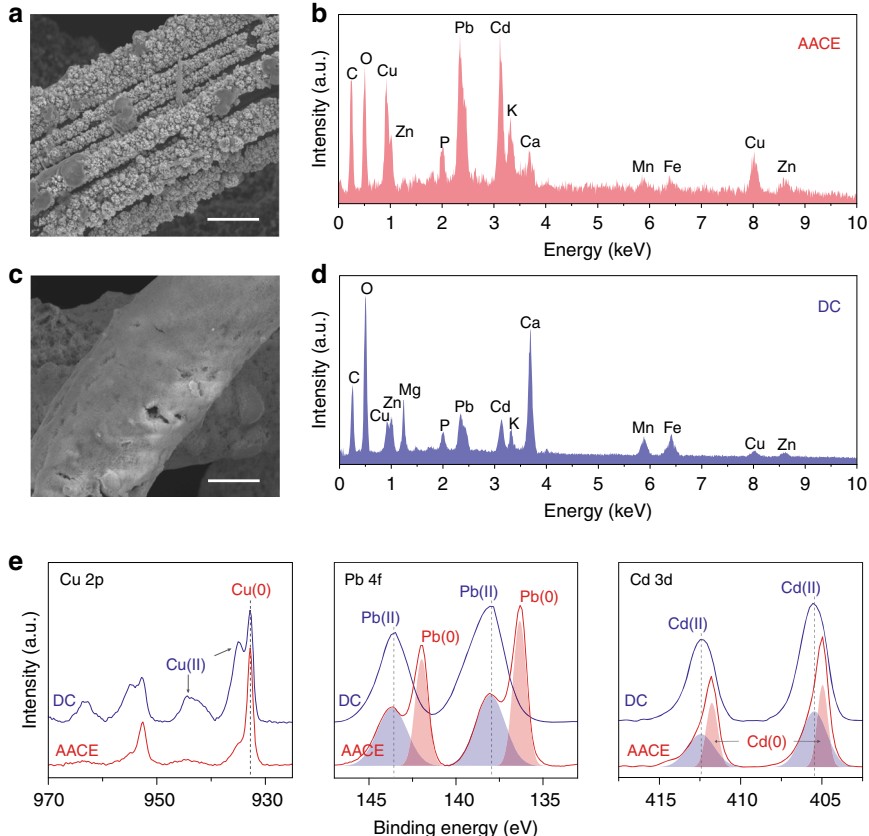

**Fig. 3** Investigation of the AACE remediation mechanism. **a** SEM image showing the morphology of the heavy metal particles extracted by the AACE method. Scale bar, 5 μm. **b** EDS showing the chemical nature of the extracted particles, where strong peaks of Cu, Pb, and Cd were found. **c** SEM image showing that the Ami-PC electrode was covered by a uniform shell when using a −10 V DC in place of the asymmetrical alternating current for the filtration. Scale bar, 5 μm. **d** EDS of the uniform shell showing a large accumulation of Ca and a small quantity of heavy metals. **e** Comparison of XPS of the heavy metal species extracted by the AACE method and the DC method

hydroxides were not conductive and prevented further removal, hence reduced the filtration efficiency in the long term. Moreover, these precipitations were loosely attached on the electrode and would be washed off after accumulation to some extent (Supplementary Fig. 7). This accounted for the large standard deviation of the DC filtration efficiency in Fig. 2f. Last, the crystal structures of Cu, Pb, and Cd particles extracted by the AACE method were examined using transmission electron microscope (TEM). The TEM images in Supplementary Fig. 8 show polycrystalline particles with the lattice spacing about 2.6, 3.5, and 2.8 Å, which correspond to (110) plane of Cu, (110) plane of Pb, and (002) plane of Cd, respectively.

**Long-term performance and plant assay**. The long-term performance of the AACE filter and the recyclability of EDTA solution were evaluated in Fig. 4a. Twenty-five equivalent soil samples were prepared by spiking with 10,000 ppm Cu (see Methods). In each cycle, the same EDTA solution was used to wash a spiked soil sample at a flow rate of 0.5 ml min$^{-1}$ followed by AACE filtration. The filtration efficiency and the mass of Cu washed out in each cycle were examined. After 25 cycles, the recycled EDTA solution had only 20% decay in its extraction ability (from ~7.5 mg to ~6 mg), illustrating that there is no accumulation of EDTA in the treated soil. Considering that our soil samples have a water retention capacity of 40%, the residue EDTA concentration in the treated soil would be 12 mmol kg$^{-1}$. The AACE filtration efficiency decreased from ~100% to ~90%

after 14 washing cycles. The SEM image in Supplementary Fig. 9a shows the morphology of the Ami-PC electrode after 14 cycles, which was covered by Cu particles with a fair number of pores blocked, indicating that the filtration efficiency decrease is mainly due to the loss of surface area and chelation sites after long-term operation. To demonstrate the regeneration of the AACE filter, 0.1 M HCl was used as the elution solution with a DC reverse bias of 1 V applied to the electrode. After the elution process, the filtration efficiency of the AACE filter returned to 100%. The SEM image in Supplementary Fig. 9b shows that all the Cu particles were recovered after elution and no damage was observed to the Ami-PC electrode.

Finally, plant assay using pea (Pisum sativum) sprouts were conducted to demonstrate the feasibility of the AACE method for agricultural land remediation (Fig. 4b, c), since dietary intake is the main route of heavy metal exposure, especially in crop vegetables planted on heavy metal contaminated soil[28]. In each pot, coir was mixed into the treated soil with a mass ratio of 1:9 to improve drainage. In the Cd positive control (soil spiked with 100 ppm Cd), most of the Cd accumulated in the root, with the median of Cd concentrations in roots, shoots, and leaves to be 210, 35, and 11 ppm, respectively. For the spiked soil sample remediated by the AACE method, the soil Cd concentration decreased to 0.25 ppm, while the concentrations of Cd in different parts of the planted pea sprouts were all below 0.1 ppm, which is the safety level for Cd in vegetables according to International Food Standards[29]. Cd accumulation in the root was not observed for the treated soil, because the residue Cd in the soil has very low

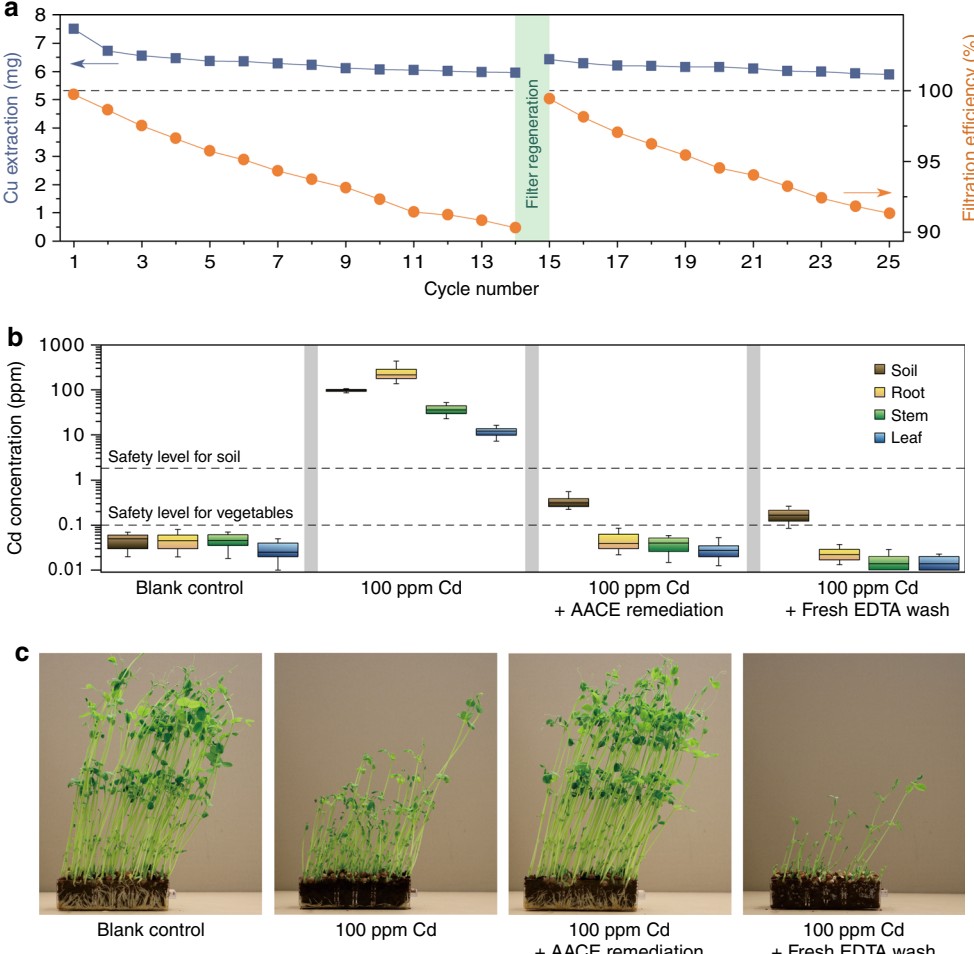

**Fig. 4** Long-term performance and plant assay. **a** Extraction ability of the recycled EDTA solution and the AACE filtration efficiency as functions of treatment cycles. Total 25 contaminated soil samples (10,000 ppm Cu) were prepared. In each cycle, the same EDTA solution was used to wash a contaminated soil sample followed by the AACE filtration, and the extraction ability of the EDTA solution was examined by the mass of Cu extracted from the soil. The AACE filter was regenerated by acid elution after the 14th cycle. **b**, **c** Plant assay using pea (Pisum sativum) sprouts. After the AACE remediation, the Cd level in the planted pea sprouts met the International Food Standards. Using fresh EDTA to wash the contaminated soil caused stunted growth of the pea sprouts. Boxplots represent median, 25th and 75th percentile. Whiskers indicate the maximum and minimum (n = 10)

bioavailability, considering that the EDTA solution cannot even leach it out. Moreover, no difference in pea growth such as shoot height and leaf size was observed between the blank control and the AACE remediated one. On the contrary, using fresh EDTA solution to wash the contaminated soil not only consumed a large amount of EDTA solution but also caused excessive soil nutrient loss, as was made evident by the stunted pea growth.

## Discussion

In summary, we developed the AACE method for the remediation of heavy metal contaminated soil, which showed a fast remediation speed and limited chemical cost. We synthesized the Ami-PC electrodes to electrochemically extract heavy metals from the soil washing effluent and demonstrated that the alternating manner of the applied voltage can dramatically enhance the extraction efficiency. We further used electron microscopy and XPS measurements to study the mechanism by comparing the heavy metal species deposited by the AACE method and the DC method. The recirculating washing system recycles the EDTA solution, which is suitable for remote-site soil remediation and high-throughput industrial operation.

Compared with conventional soil washing method using fresh EDTA solution, the AACE method produced no secondary pollution, and no obvious soil degradation was observed after the treatment. In addition, the remarkable stability and regeneration properties of the AACE filter after long-term operation make us envision the use of this platform for the recovery of heavy metals from the waste streams in various manufacturing and chemical industries. Further optimization of the operation system and the applied voltage could also enhance its scalability.

## Methods

**Ami-PC electrode fabrication**. Carbon felt (Alfa Aesar, 3.18 mm thick) was cut into circular pieces with a diameter of 1 cm. Polyacrylonitrile (PAN, Sigma-Aldrich, molecular weight ~150,000), Super P carbon black (Alfa Aesar, 99%) and N,N-dimethylformamide (DMF, Fisher Chemical, 99.9%) were mixed with a mass ratio of 1:1:30 and stirred overnight to form a uniform slurry. The circular carbon felt pieces were dip-coated with the slurry and air-dried on a hotplate (70 °C). The coated carbon felt pieces were put into a 25 ml water bath; 80 mg ml$^{-1}$ hydroxylamine hydrochloride (Sigma-Aldrich, 99%) and 60 mg ml$^{-1}$ sodium carbonate (Fisher Chemical, 99.5%) were added. The water bath was kept at 70 °C for 90 min. Then the carbon felt pieces were washed with deionized water and dried in a vacuum furnace (80 °C).

**Characterization of soil properties**. For measuring the metal concentrations, the soil samples were acid digested in accordance with U.S. EPA Method 3050 and analysed using an inductively coupled plasma mass spectrometry (ICP-MS). Soil pH was measured using a benchtop pH meter with a soil-water volume ratio of 1:2. The soil texture (the relative distribution of sand, silt, and clay) was determined using a standard hydrometer method. The organic carbon content was measured by the Walkley–Black method. The cation exchange capacity was determined by saturating the exchangeable sites with sodium ions followed by substitution with magnesium ions.

**Soil remediation experiment**. In each remediation experiment, 3 g of contaminated soil was put into a 3-cm-long plastic tube with a 1-cm inner diameter. The packed soil column has a bulk density of 1.27 g cm$^{-3}$. Two pieces of tissue papers were put on each side of the soil column to prevent soil particles from being washed out. An electrochemical filtration device composed of two Ami-PC electrodes with a tissue paper between as a separator was put at the end of the plastic tube and connected to a power supply. The asymmetrical alternating current was generated by GIGOL DG1022A. The direct current was provided by Keithley 2400. In the first treatment cycle, 8 ml of 30 mM EDTA disodium salt (Sigma-Aldrich, 98.5–101.5%) solution was infused by a syringe pump to wash through the soil column and then filtrated by the AACE filter (Supplementary Fig. 10a). In total, 1.2 ml of the solution was retained by the soil column to make it saturated, because the packed soil column has a porosity of 51%. In total, 0.8 ml of the solution was retained by the porous Ami-PC electrodes of the AACE filter. In the following treatment cycles, the rest 6 ml of the EDTA solution was circulated by a peristaltic pump to wash through the soil column and the AACE filter repeatedly (Supplementary Fig. 10b).

**Material characterization**. The FTIR spectrum was measured using Nicolet iS50 in the attenuated total reflectance mode. The SEM images and EDS were taken by FEI Nova NanoSEM 450 with an acceleration voltage of 5 kV for imaging and 15 kV for EDX collection. The XPS spectra were collected using PHI VersaProbe Scanning XPS Microprobe with an Al (Kα) source. The TEM images were taken by FEI Titan TEM at 300 keV. For TEM analysis, three soil samples spiked with 1000 ppm Cu, 1000 ppm Pb, and 1000 ppm Cd, respectively, were treated by the AACE method. The working electrodes were ultra-sonicated in 15 ml water for 20 min. TEM grids were then promptly dipped in the suspensions and allowed to dry.

**Long-term performance evaluation**. Twenty-five equivalent soil samples were prepared by mixing 3 g of 10,000 ppm Cu contaminated soil with 1.2 ml of 30 mM EDTA disodium salt solution. The 1.2 ml EDTA solution was added to make the soil sample saturated. For the first washing cycle, 6.8 ml of 30 mM EDTA solution was infused by a syringe pump to wash a spiked soil sample at a flow rate of 0.5 ml min$^{-1}$ and then filtrated by the AACE filter. In total, 0.8 ml of the EDTA solution was retained by the porous Ami-PC electrodes of the AACE filter. The filtration generated 6 ml of washing effluent, which was collected and infused by the syringe pump to wash other spiked soil samples in the following washing cycles.

**Plant assay**. Pisum Sativum (pea) seeds were purchased from Window Garden LLC. In each pot, coir (Root Naturally LLC) and the treated soil samples were mixed at mass ratio of 1:9 to improve drainage. After 15 days from planting, the pea sprouts were washed with deionized water and cut into leaves, stems, and roots, which were then air-dried at 70 °C for 24 h, weighted for dry matter yield, and ground. Concentrations of heavy metals were determined after digestion using U.S. EPA Method 3050. The digestives were analyzed for Cd by ICP-MS.

## Data availability

The authors declare that the main data supporting the findings of this study are available within the article and its Supplementary Information files. Extra data are available from the corresponding author upon reasonable request.

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

## Author contributions

J.X. and Y.C. conceived the idea. J.X. conducted the experiments. C.L. and T.W. helped with the synthesis of the Ami-PC electrodes. C.L. conducted the EDS characterization. P.-C.H. conducted the FTIR characterization. J.Z. conducted the TEM characterization. J.T. helped with the pea planting. K.L. helped with the data analysis. J.X. and Y.C. analysed the data and co-wrote the paper. All authors commented on the paper.

## Additional information

**Competing interests:** The authors declare no competing interests.

