## [Peer Review File · Nature Communications]

Reviewers' comments:

Reviewer #1 (Remarks to the Author):

This paper provides a new idea about how to improve the remediation efficiency for heavy metal contaminated soil, by using asymmetrical alternating current electrochemistry. The results seem to be promising, and the soils after remediation keep good quality and are satisfied with plant growth. But, I have following comments:

1. The soil properties are unclear. You mentioned that when you prepared the heavy metal spiked soils, you treated them at 70 °C. This is unsuitable, because heating the soil may result in the change of soil properties from that in field.
2. The soil column is very small with a length of 3 cm, and diameter of 1 cm. It is very difficult to scale-up to a much more big column.
It is also surprising that the soil metal concentration decreased so quickly within several hours. It also means that the heavy metals in the soil have very high mobility or bioavailability, which is not widely seen in practice soil.
3. EDTA leaching the soil got a rapid removal of metals from the soils. Generally, to leach the soil can lead to the blocking of soil pores. How do you avoid such blocking?
4. You said that the amidexime could compete with EDTA for the complexed metal. But, you should provide the data, such as complexing constant, about the complex of metal with the amidexime. In addition to that, you should have a mass balance about the metals. Was the deposited metal on electrode surface equal to the removal of metal from the soil?
5. I think that an additional experiment using water or salt solution to leach the soil instead of EDTA should be added. I wonder that such leaching can also get good result, because the metal in the soil has very good solubility.
6. Regarding the plant experiment, the plant can well grow in the soil after AACE. I suggest to add some data about the residue of EDTA in the soil. The Cd concentrations in the different tissues of plant in this paper seem to be lower than those in references with the similar soil Cd.

Reviewer #2 (Remarks to the Author):

This manuscript presents interesting and fruitful results on contaminated soil remediation with the utilization of proposed method - AACE. The topic is of interests to the readers from multiple disciplines and the work conducted in this manuscript is of novelty. It could be considered for publication after some specific comments are addressed.

The information presented in Table S1 it is not so accurate. For example, the chemical used for soil washing can be regenerated as well. The regeneration can be conducted separately. The nutrition could be retained with the washing solution reused. The energy consumption is not convincing.

It is not stated about the criteria for selecting the characteristic pollutants as well as the setting of their content in soil, since only Cu Pb and Cd are investigated.

The type and composition of soil will affect the remediation effect significantly, especially the organic substances. The characteristics of the soil used has not been introduced.

Regarding the chelating agent solution is recycled and reused, whether the impacts from the enrichment of water-soluble metals towards the soil or the solution been investigated?

What's the reason for the bad performance of soil using fresh EDTA to wash (as presented in fig 4).

Point-by-point response to the reviewers' comments (in blue)

Reviewer #1:

This paper provides a new idea about how to improve the remediation efficiency for heavy metal contaminated soil, by using asymmetrical alternating current electrochemistry. The results seem to be promising, and the soils after remediation keep good quality and are satisfied with plant growth. But, I have following comments:

Response:

We are glad that Reviewer#1 appreciate the novelty and potential impact of our idea here.

Comment 1

1. The soil properties are unclear. You mentioned that when you prepared the heavy metal spiked soils, you treated them at 70 °C. This is unsuitable, because heating the soil may result in the change of soil properties from that in field.

Response:

We thank the reviewer for this comment, and we would like to mention that drying at 70 °C is commonly used when preparing spiked soil samples. The purpose of such drying process is to fully remove the moisture thus to get a reliable measurement of the dry mass of the soil samples, which is very important for quantitatively controlling the heavy metal concentrations. We also want to mention that we further aged the soil at 80 °C for one month, in order to decrease the solubility and mobility of the spiked heavy metals. The soil sample used in the blank control of plant assay also underwent the drying and the aging process, and the healthy growth of pea sprouts corroborates the minimal effect of these processes.

To address the reviewer's point, we measured several soil properties, including soil texture, organic carbon content, pH and cation exchange capacity of the fresh soil samples and the aged soil samples. The methods used for these measurements are added in the revised Method, and the results are presented in the new Supplementary Table 1 (also shown below). From the results, we can conclude that both the drying and the aging process didn't change the soil properties. Such discussion is now added in the revised manuscript and is also listed below for the reviewer's convenience.

	Sand (%)	Silt (%)	Clay (%)	pH	OC (g/kg)	CEC (cmol _c /kg)
Fresh soil	45	37	18	6.2	28.4	17.9
Aged soil	45	39	16	6.8	25.5	18.3

Supplementary Table 1. Textural analysis and physicochemical properties of the fresh and aged soil.

OC, organic carbon. CEC, cation exchange capacity.

(Page 6 Line 12-22): *“To quantitatively evaluate the remediation performance of the AACE method, a series of experiments using synthetically contaminated soil were conducted. The soil used in this study was collected from the O'Donohue Family Stanford Educational Farm. The soil was then air-dried at 70 °C, and passed through a 2-mm sieve to remove coarse fragments. Contaminated soil samples with different heavy metal (Cu, Pb, Cd) concentrations were synthetically prepared by thoroughly mixing the*

clean soil samples with the corresponding nitrate-salt solutions. After the spiking process, the contaminated soil samples were air-dried and aged for one month at 80 °C, in order to simulate practical polluted soil by decreasing the solubility and mobility of the heavy metals (Supplementary Fig. 5). Characteristics of the fresh and the aged soil samples, including soil texture, organic carbon, pH, and cation exchange capacity are provided in Supplementary Table 1, corroborating that the aging process didn't change the soil properties from that in field.”

Comment 2

2.1. The soil column is very small with a length of 3 cm, and diameter of 1 cm. It is very difficult to scale-up to a much more big column.

Response:

We agree with the reviewer that scale-up is important for practical treatment of polluted soil, which is a system-level optimization. This work aims to demonstrate the new concept of alternating current electrochemistry and its feasibility for removing heavy metals from the EDTA soil washing solution. With the understanding on the extraction mechanism, we made a filter device and achieved fast and efficient treatment. Albeit the diameter of the filter in our device is only 1 cm, this method can be easily scaled up by running multiple filters in parallel. The main reason why we use small filters in our experiments is to have a more controlled and uniform flow of the EDTA solution, considering that the heterogeneity of a large soil column would introduce additional fluidic complexity. Therefore, demonstrating the feasibility of our AACE method on a smaller device is preferable.

2.2. It is also surprising that the soil metal concentration decreased so quickly within several hours. It also means that the heavy metals in the soil have very high mobility or bioavailability, which is not widely seen in practice soil.

Response:

We also agree with the reviewer that the mobility and solubility of heavy metals in practical polluted soil is usually low. Therefore, we had an aging process when preparing the contaminated soil samples. “After the spiking process, the contaminated soil samples were air-dried and aged for one month at 80 °C.” We also did some new experiments to show that such aging process greatly decreased the mobility and solubility of the heavy metals. The data and more discussions are provided in our response to the reviewer’s comment 5.

Comment 3

3. EDTA leaching the soil got a rapid removal of metals from the soils. Generally, to leach the soil can lead to the blocking of soil pores. How do you avoid such blocking?

Response:

We thank the reviewer for this comment, and we agree that EDTA leaching will lead to the blocking of soil pores for in-situ soil flushing. However, in the application scenario of our AACE method, the polluted soil is excavated from the contaminated site, get treated in the device, and then discharged back. Therefore, post-treatment such as mixing coarse sands into the treated soil can be easily done to improve the drainage for plant growth.

In our experiments, the soil samples were passed through a 2-mm sieve to remove coarse fragments as a pretreatment to make the treatment more efficient and our experiments more repeatable, because large

coarse sand particles can consist a large mass proportion of the soil sample while absorb very little amount of heavy metals. Consequently, the treated soil samples have very poor drainage and will cause plant drowning. To address this issue, we mixed coir into the treated soil with a mass ratio of 1:9 to improve drainage for the plant assay. Such information is now added in the revised manuscript and is also listed below for the reviewer’s convenience.

(Page 6 Line 14): “The soil was then air-dried at 70 °C, and passed through a 2-mm sieve to remove coarse fragments.”

(Page 10 Line 21): “In each pot, coir was mixed into the treated soil with a mass ratio of 1:9 to improve drainage.”

Comment 4

4.1. You said that the amidoxime could compete with EDTA for the complexed metal. But, you should provide the data, such as complexing constant, about the complex of metal with the amidoxime.

Response:

Per the reviewer’s request, we measured the adsorption isotherms of heavy metals on our Ami-PC electrode. The results are added as the new Supplementary Fig. 2 (also shown below). Fig. A shows the equilibrium adsorption amounts (q_e) of Cu, Pb and Cd ions onto the Ami-PC electrode after the equilibrium time (6 h) as a function of equilibrium concentration (C_e) in the solution. Fig. B shows the linearized format of the isotherms, which fits very well with Langmuir model. Fig. C shows the maximum adsorption capacity (q_m) and the Langmuir adsorption constant (K_L), calculated from the slope and the intercept in Fig. B. The high adsorption capacity indicates the strong chelation sites provided by the amidoxime coating.

Supplementary Fig. 2. Adsorption isotherms of heavy metals on the Ami-PC electrode. (A) The equilibrium adsorption amounts (q_e) of Cu, Pb and Cd ions onto the Ami-PC electrode after the equilibrium time (6 h) as a function of equilibrium concentration (C_e) in the solution, fitted by the Langmuir model (equation 1). (B) The adsorption isotherms displayed in their linearized format (equation 2). (C) The Langmuir isotherm parameters are calculated from the slope and intercept of the linear fitting in B. The q_m and K_L represent the maximum adsorption capacity of the Ami-PC electrode and the Langmuir adsorption constant. The high adsorption capacity indicates the strong chelation sites provided by the amidoxime coating.

$$q_e = \frac{q_m K_L C_e}{1 + K_L C_e} \quad (\text{equation 1})$$

$$\frac{C_e}{q_e} = \frac{1}{q_m K_L} + \frac{C_e}{q_m} \quad (\text{equation 2})$$

4.2. In addition to that, you should have a mass balance about the metals. Was the deposited metal on electrode surface equal to the removal of metal from the soil?

Response:

We thank the reviewer for this comment, and we want to mention that we have measured the AACE filtration efficiency in Fig. 2D and Fig. 4A. The AACE filtration efficiency is defined as the percentage of the heavy metal cations electrodeposited by the AACE filter from the washing effluent, and is determined by measuring the heavy metal concentrations in the washing effluent before and after passing through the AACE filter at a specific flow rate. According to the conservation of mass, the heavy metals removed from the washing effluent would be retained by the filter. The EDTA was thus recycled and used to wash the contaminated soil repeatedly. Therefore, the removal of heavy metals from the soil was deposited on the electrode rather than accumulated in the washing solution.

Comment 5

5. I think that an additional experiment using water or salt solution to leach the soil instead of EDTA should be added. I wonder that such leaching can also get good result, because the metal in the soil has very good solubility.

Response:

Per the reviewer’s request, we used deionized water to wash the contaminated soil (1,000 ppm Pb). The results are added as the new Supplementary Fig. 5 (also shown below). For the unaged soil (spiked with Pb(NO₃)₂ solution without post-treatment), washing with pure water can leach out a large proportion (~80%) of the spiked Pb. However, for the aged soil, pure water can only leach out 20% of the Pb after 3 hours, which is much less compared with our AACE method and the fresh EDTA solution wash (Fig. 2B). This result corroborates that the aging process enabled the heavy metals to adsorb to the soil particles and decreased their solubility and mobility.

Supplementary Fig. 5. Washing the contaminated soil (1,000 ppm Pb) with deionized water. For the unaged soil (spiked with Pb(NO₃)₂ solution without post-treatment), washing with pure water can leach out a large proportion (~80%) of the spiked Pb. However, for the aged soil, pure water can only leach out

20% of the Pb after 3 hours, which is much less compared with our AACE method and the fresh EDTA solution wash (Fig. 2B). This result corroborates that the aging process enabled the heavy metals to adsorb to the soil particles and decreased their solubility and mobility.

Comment 6

6.1. Regarding the plant experiment, the plant can well grow in the soil after AACE. I suggest to add some data about the residue of EDTA in the soil.

Response:

We thank the reviewer for this comment, and we want to mention that it's very hard to extract the residue EDTA from soil completely without loss. Therefore, direct measurement would underestimate the concentration. However, from the long-term experiments shown in Fig. 4A, we can see that the EDTA solution kept a stable extraction ability after cycles of experiments, indicating that there is no accumulation of EDTA in the treated soil. Because our soil samples have a water retention capacity of 40%, there will be 2 ml of EDTA solution retained after treating 3 g of dry soil. Consequently, the residue EDTA concentration would be 12 mmol/kg in the treated soil. Although we didn't see any disadvantageous effect of the residue EDTA on the pea sprouts growth, the residue EDTA can be easily washed out by water if necessary for other future applications. Such discussion is now added in the revised manuscript and is also listed below for the reviewer's convenience.

(Page 10 Line 5-8): "After 25 cycles, the recycled EDTA solution had only 20 percent decay in its extraction ability (from ~7.5 mg to ~6 mg), illustrating that there is no accumulation of EDTA in the treated soil. Considering that our soil samples have a water retention capacity of 40%, the residue EDTA concentration in the treated soil would be 12 mmol/kg."

6.2. The Cd concentrations in the different tissues of plant in this paper seem to be lower than those in references with the similar soil Cd.

Response:

We thank the reviewer for this comment. For the pea sprouts growing in the 100 ppm Cd contaminated soil, Cd mainly accumulated in the root, reaching a concentration of double of the soil Cd concentration. However, for the treated soil (both by AACE and by fresh EDTA washing), the concentration of Cd in the different tissues of plant are lower than the residue Cd concentration in the treated soil. This is because the residue Cd in the treated soil has very low bioavailability, considering that it cannot even be leached out by the EDTA solution. Such discussion is now added in the revised manuscript and is also listed below for the reviewer's convenience.

(Page 11 Line 3-5): "Cd accumulation in the root was not observed for the treated soil, because the residue Cd in the soil has very low bioavailability, considering that the EDTA solution cannot even leach it out."

Reviewer #2:

This manuscript presents interesting and fruitful results on contaminated soil remediation with the utilization of proposed method - AACE. The topic is of interests to the readers from multiple disciplines and the work conducted in this manuscript is of novelty. It could be considered for publication after some specific comments are addressed.

Response:

We appreciate Reviewer#2's understanding of the interesting results of our work.

Comment 1

1. The information presented in Table S1 it is not so accurate. For example, the chemical used for soil washing can be regenerated as well. The regeneration can be conducted separately. The nutrition could be retained with the washing solution reused. The energy consumption is not convincing.

Response:

We thank the reviewer for this comment, and we have deleted the Table S1. In addition, we rewrote the conclusion part (also shown below).

(Page 11 Line 12-22 and Page 12 Line 1,2): "In summary, we developed the AACE method for the remediation of heavy metal contaminated soil, which showed a fast remediation speed and limited chemical cost. We synthesized the Ami-PC electrodes to electrochemically extract heavy metals from the soil washing effluent and demonstrated that the alternating manner of the applied voltage can dramatically enhance the extraction efficiency. We further used electron microscopy and XPS measurements to study the mechanism by comparing the heavy metal species deposited by the AACE method and the DC method. The recirculating washing system recycles the EDTA solution, which is suitable for remote-site soil remediation and high-throughput industrial operation. Compared with conventional soil washing method using fresh EDTA solution, the AACE method produced no secondary pollution, and no obvious soil degradation was observed after the treatment. In addition, the remarkable stability and regeneration properties of the AACE filter after long-term operation make us envision the use of this platform for the recovery of heavy metals from the waste streams in various manufacturing and chemical industries. Further optimization of the operation system and the applied voltage could also enhance its scalability."

Comment 2

2. It is not stated about the criteria for selecting the characteristic pollutants as well as the setting of their content in soil, since only Cu Pb and Cd are investigated.

Response:

We thank the reviewer for this comment, and we agree that there are dozens of elements in the family of heavy metals, considering that a density of more than 5 g/cm³ is often mentioned as the defining factor. However, Cu, Pb and Cd are the most commonly found contaminants in polluted soil and have resulted in a large number of widespread poisoning incidents. Therefore, the treatment of Cu, Pb and Cd contaminations attracts most attention and is widely used to evaluate the performance of a treatment method in literatures.

Another reason for choosing Cu, Pb and Cd is that we want to demonstrate the versatility of the AACE method by treating three different heavy metals with different toxicities and different concentrations. Cu is an essential substance to human life, and it only causes diseases at high exposure. On the contrary, only small amount of Cd can cause renal disfunction. The toxicity of Pb is in the middle between Cu and Cd. Therefore, there is an order of magnitude difference among the safety levels for these three heavy metals, and it would be inappropriate to treat them from the same beginning concentrations. The setting of their content is thus according to their toxicity and their typical concentrations found in contaminated sites. Such discussion is now added in the revised manuscript and is also listed below for the reviewer's convenience.

(Page 7 Line 1-4): "Considering the large variation in hazardous level among different contaminated sites (26) and the disparate safety standards for different heavy metals (27), three synthetically contaminated soil samples were prepared by spiking with 10,000 ppm Cu, 1,000 ppm Pb and 100 ppm Cd, respectively. The setting of their content is according to their toxicity and their typical concentrations found in contaminated sites."

Comment 3

3. The type and composition of soil will affect the remediation effect significantly, especially the organic substances. The characteristics of the soil used has not been introduced.

Response:

Per the reviewer's request, we measured several soil properties, including soil texture, organic carbon content, pH and cation exchange capacity, of the fresh soil samples and the aged soil samples. Soil pH was measured using a benchtop pH meter with a soil-water volume ratio of 1:2. The soil texture (the relative distribution of sand, silt and clay) was determined using a standard hydrometer method. The organic carbon content was measured by the Walkley-Black method. The cation exchange capacity was determined by saturating the exchangeable sites with sodium ions followed by substitution with magnesium ions. The results are presented in the new Supplementary Table 1 (also shown below).

	Sand (%)	Silt (%)	Clay (%)	pH	OC (g/kg)	CEC (cmol _c /kg)
Fresh soil	45	37	18	6.2	28.4	17.9
Aged soil	45	39	16	6.8	25.5	18.3

Supplementary Table 1. Textural analysis and physicochemical properties of the fresh and aged soil. OC, organic carbon. CEC, cation exchange capacity.

Comment 4

4. Regarding the chelating agent solution is recycled and reused, whether the impacts from the enrichment of water-soluble metals towards the soil or the solution been investigated?

Response:

We thank the reviewer for this comment, and we want to mention that the enrichment behavior of metal ions depends on the metal species. For the heavy metals, they were washed out from the soil by the EDTA washing solution, and then they were immediately extracted out from the washing effluent by the AACE

filter. Therefore, all the heavy metals were enriched on the working electrode of the AACE filter. Such accumulated heavy metals can then be recovered by acid elution (Supplementary Fig. 9).

For other nutrient metal ions (Na^+ , Mg^{2+} , etc.), they can also be washed out by the EDTA solution. However, they cannot be extracted out by the AACE filter because of their lower standard reduction potential. After the first washing cycle, the nutrient metal concentration in the EDTA solution established an equilibrium with the nutrient metal concentration in the soil. Consequently, when we used the recycled EDTA solution to wash the soil in the following cycles, it didn't wash out more nutrient metal ions (orange curve in Fig. 2E). However, excessive nutrient loss happened when using fresh EDTA to wash the soil (blue curve in Fig. 2E). Therefore, for nutrient metals, there was no enrichment in the AACE filter nor the EDTA solution. The nutrient metals were kept in the soil. Such discussion is now added in the revised manuscript and is also listed below for the reviewer's convenience.

(Page 7 Line 18-23, and Page 8 Line 1,2): "The concentration of magnesium in soil was also monitored during three different treatment methods at a flow rate of 0.1 ml/min (Fig. 2E). For nutrient metal ions (Na^+ , Mg^{2+} , etc.), they can also be washed out by the EDTA solution. However, they cannot be extracted out by the AACE filter because of their lower standard reduction potential. After the first washing cycle, the nutrient metal concentration in the EDTA solution established an equilibrium with the nutrient metal concentration in the soil. Consequently, when we used the recycled EDTA solution to wash the soil in the following cycles, it didn't wash out more nutrient metal ions. However, excessive nutrient loss happened when using fresh EDTA to wash the soil."

Comment 5

5. What's the reason for the bad performance of soil using fresh EDTA to wash (as presented in fig 4).

Response:

We thank the reviewer for this comment. This is due to the excessive soil nutrient loss caused by the fresh EDTA wash. As shown in Fig. 2E and also as mentioned in our response to the reviewer's comment 4, using fresh EDTA to wash the soil will leach out all the soluble metal ions, including those nutrient metals that are necessary for plant growth. Consequently, the pea growth was stunted by nutrient deficiency.

Reviewers' comments:

Reviewer #1 (Remarks to the Author):

The authors revised their paper following the suggestion or comments from the reviewers. I have one more question:

It is unlike other electrokinetic remediation of heavy metal contaminated soil. The main contribution is that the authors constructed a modified electrode, which can attract metal-EDTA anions, and then reduced them to zero-valent metal and released recyclable EDTA for further cycling. So, to drive the metal-EDTA complex from soil column is strongly based on water power, not electromigration or electroosmosis. But, this is not mentioned how you applied water power to the soil column. In addition to that, how much is the soil density packed in soil column? Do you calculate the soil pore volume, which maybe useful to explain why the metal in soil column is rapidly reduced with very short term.

Reviewer #2 (Remarks to the Author):

The revised manuscript have addressed my previous comments well.

Point-by-point response to the reviewers' comments (in blue)

Reviewer #1:

The authors revised their paper following the suggestion or comments from the reviewers. I have one more question: It is unlike other electrokinetic remediation of heavy metal contaminated soil. The main contribution is that the authors constructed a modified electrode, which can attract metal-EDTA anions, and then reduced them to zero-valent metal and released recyclable EDTA for further cycling.

Response:

We are glad that Reviewer#1 appreciate the contribution of our manuscript.

Comment 1

So, to drive the metal-EDTA complex from soil column is strongly based on water power, not electromigration or electroosmosis. But, this is not mentioned how you applied water power to the soil column.

Response:

We thank the reviewer for this comment, and we have added a more detailed description on how we applied water power to the soil column in the Methods part of our manuscript and a new Supplementary Fig. 10 to illustrate the experimental setup. Specifically, we used a syringe pump to infuse the EDTA solution to wash through the soil column and the AACE filter. After the first soil washing cycle, we replaced the syringe pump with a peristaltic pump, which circulated the EDTA solution to wash the soil column and the AACE filter repeatedly. Such information is now added in the revised manuscript and is also listed below for the reviewer's convenience.

(Page 20 Line 4-20): "In the first treatment cycle, 8 ml of 30 mM EDTA disodium salt (Sigma-Aldrich, 98.5-101.5%) solution was infused by a syringe pump to wash through the soil column and then filtrated by the AACE filter (Supplementary Fig.10 A). 1.2 ml of the solution was retained by the soil column to make it saturated, because the packed soil column has a porosity of 51%. 0.8 ml of the solution was retained by the porous Ami-PC electrodes of the AACE filter. In the following treatment cycles, the rest 6 ml of the EDTA solution was circulated by a peristaltic pump to wash through the soil column and the AACE filter repeatedly (Supplementary Fig.10 B)."

Supplementary Fig. 10. Illustration of the experimental setup. (A) A syringe pump was first used to infuse the soil washing solution, which washed through the soil column and the AACE filter, and accumulated above the AACE filter. (B) After all the soil washing solution went through the soil column for the first time, a peristaltic pump was used to circulate the solution to wash through the soil column repeatedly.

Comment 2

In addition to that, how much is the soil density packed in soil column?

Response:

In each remediation experiment, 3 g of contaminated soil was put into a 3 cm long plastic tube with a 1 cm inner diameter. Therefore, the packed soil column has a bulk density of 1.27 g/cm³. Per the reviewer's request, such information is now added in the revised manuscript and is also listed below for the reviewer's convenience.

(Page 19 Line 22): "The packed soil column has a bulk density of 1.27 g/cm³."

Comment 3

Do you calculate the soil pore volume, which may be useful to explain why the metal in soil column is rapidly reduced with very short term?

Response:

We thank the reviewer for this comment, and we agree that a large porosity of the soil column can enhance the leaching out speed of heavy metal ions. During our experiments, we found that after the soil washing solution flowed through the packed soil column, 1.2 ml of the solution would be retained. The total volume of the soil column is 2.36 cm³. Therefore, the porosity can be calculated out to be 51%. Such information is now added in the revised manuscript and is also listed below for the reviewer's convenience.

(Page 20 Line 6,7): "1.2 ml of the solution was retained by the soil column to make it saturated, because the packed soil column has a porosity of 51%."

Reviewer #2:

The revised manuscript has addressed my previous comments well.

Response:

We thank the Reviewer#2 again for his/her constructive advice.

REVIEWERS' COMMENTS:

Reviewer #1 (Remarks to the Author):

I am satisfied with the answers and revisions to my questions. Thus, I recommend to publish the paper without additional revision.

Reviewer #2 (Remarks to the Author):

The authors has made significant revision works based on reviewers' comments. From my side, I don't have any more comments for this paper.